# A screening system for identifying interacting proteins using biomolecular fluorescence complementation and transposon gene trap

**Honami Miyakura[1], Mei Fukuda[1], Hiroya Enomoto[1], Kosuke Ishikawa[2]\*, Shinya Watanabe[3], Kentaro Semba[1,3]**

**1** Department of Life Science and Medical Bioscience, School of Advanced Science and Engineering, Waseda University, Shinjuku-ku, Tokyo, Japan, **2** Laboratory of Japan Biological Informatics Consortium (JBiC), Koto-ku, Tokyo, Japan, **3** Translational Research Center, Fukushima Medical University, Fukushima, Japan

\* ishikawakosuke@gmail.com

**Data Availability Statement:** All relevant data are within the manuscript and its Supporting Information files.

## Abstract

We have established a new screening system for identifying interacting proteins by combining biomolecular fluorescence complementation (BiFC) and a transposon gene trap system. This system requires creation of a bait strain that stably expresses a fusion product of part of the fluorescent monomeric Kusabira-Green (mKG) protein to a protein of interest. A *PiggyBac* transposon vector is then introduced into this strain, and a sequence encoding the remainder of mKG is inserted into the genome and fused randomly with endogenous genes. The binding partner can be identified by isolating cells that fluoresce when BiFC occurs. Using this system, we screened for interactors of p65 (also known as RELA), an NF-κB sub-unit, and isolated a number of mKG-positive clones. 5′- or 3′-RACE to produce cDNAs encoding mKG-fragment fusion genes and subsequent reconstitution assay identified PKM, HSP90AB1, ANXA2, HSPA8, and CACYBP as p65 interactors. All of these, with the exception of CACYBP, are known regulators of NF-κB. Immunoprecipitation assay confirmed endogenously expressed CACYBP and p65 formed a complex. A reporter assay revealed that CACYBP enhanced 3κB reporter activation under TNFα stimulation. This screening system therefore represents a valuable method for identifying interacting factors that have not been identified by other methods.

## Introduction

Elucidating protein–protein interactions is critical for analyzing protein function and identifying potential therapeutic targets. However, existing methods for identifying such interactions are limited in their utility. For example, the commonly used two-hybrid method only recognizes interactions in the nucleus and can generate false positives if the proteins of interest have transcriptional activity. Fluorescence resonance energy transfer (FRET) can be used to probe the dynamics of protein interactions in living cells. However, the visualization is real-time, the fluorescence does not accumulate, and requires the use of highly sensitive detection devices.

**Funding:** The authors received no specific funding for this work.

**Competing interests:** The authors have declared that no competing interests exist.

Also, a background of fluorescent proteins is inevitable. The utility of mass spectrometry (MS) is also limited because one cannot know whether the interaction is direct or indirect, and high expression levels are required.

Here, we focused on Bimolecular Fluorescence Complementation (BiFC), a highly sensitive and simple technology that is not subject to the weaknesses listed above. BiFC occurs when two complementary fragments of a fluorescent protein, neither of which fluoresce on their own, come into proximity and form a native-like structure that reconstitutes fluorescence. By fusing two proteins of interest, one to each fragment of the fluorescent protein, the level of interaction between the proteins of interest can be evaluated by the degree of fluorescence [1, 2]. Although it may take longer for fluorescence to mature than in the case of the FRET system, the fluorescence accumulates and become stable [3], allowing for the detection of even weak interactions.

In order to screen for binding partners of a bait protein, it is necessary to prepare a library of prey proteins. We used a DNA transposon to achieve library creation and screening simultaneously [4, 5]. The DNA transposon can insert any DNA element into the genome [6], and one can easily obtain a cell library in which a fragment of the fluorescent protein is fused with a different gene in each cell. Compared to viral vector systems, the transposon system achieves greater randomness of insertion, is easier to use, and can incorporate sequences that would hinder viral packaging [7]. In this study, we used the hyperactive *PiggyBac* transposon for highly efficient genomic integration [8].

As a case study, we screened for novel protein interactors of p65, a component of the transcription factor NF-κB that forms a heterodimer with another NF-κB component such as p50 (also known as NFKB1). p65 plays important roles in the transcriptional regulation of immune responses and has attracted attention as a target for treatment of numerous diseases [9–14]. Details of the screening system, methods for identifying candidate genes, and their effects on NF-κB activity are described.

## Materials and methods

### BiFC kit

For BiFC screening system, we used the Coral Hue® Fluo-chase kit (MBL). This kit includes p65-mKGN and p50-mKGC, in which the N-terminal and C-terminal fragments of the fluorescent protein mKG (monomeric Kusabira Green) are fused to fragments of NF-κB components p65 and p50. When the complementary components are co-expressed, the chromophore of mKG is reconstituted and fluoresces. The bait strain used in the screening system of this study stably expressed p65-mKGN. The nucleotide and amino acid sequences of p65-mKGN and p50-mKGC are shown in S1 Appendix of S1 File.

### Cell culture

HeLa and HEK293T were maintained in DMEM (Fujifilm WAKO, Tokyo, Japan) supplemented with 10% FBS (Nichirei Biosciences, Tokyo, Japan), 100 μg/mL streptomycin sulfate, 100 U/mL penicillin G potassium (SMPG) at 37°C under 5% $CO_2$.

### Bait cell line

After infecting HeLa cells with a retroviral expression vector (pMXs-p65-mKGN-IRES-bla[R]), the cells were selected in bulk with blasticidin and then single cell sorting was performed using a cell sorter SH800Z (Sony, Tokyo, Japan). A cell clone that was able to withstand repeated

single cell sorting, had high protein expression of p65-mKGN, and high BiFC fluorescence with the introduction of p50-mKGC was selected. The cell line was named HeLa-p65-mKGN.

## Screening of p65 interaction proteins by FACS

HeLa-p65-mKGN cells were seeded in a 12-well plate at $8 \times 10^4$ cells/well. After overnight culture, transposon donor vectors shown in Fig 1(A) were co-introduced using polyethyleneimine (PEI, Polysciences) with the hyPBase transposase expression vector. The DNA solution used for transfection was adjusted to 0.75 µg donor vector and 0.25 µg transposase (donor: transposase = 3:1). In the case of a vector having a splicing acceptor on the 5′ side of mKGC, it is expected that the N-terminal side of mKGC would fuse to the exon sequence of a gene and expressed. In the case of a vector having a splicing donor (SD) sequence on the 3′ side, it is expected that the C-terminal side of mKGC would fuse to the exon sequence of a gene. One day after transfection, the cells were passaged into a 10-cm dish/sample, and 7 days after transfection, mKG-positive cells were collected in bulk using a cell sorter. After propagation and passage to a 10-cm dish, cloning was performed by single cell sorting. After confirming the fluorescence of the clone with a flow cytometer, the expression of the mKGC fusion protein was confirmed by immunoblot analysis using an anti-mKGC antibody.

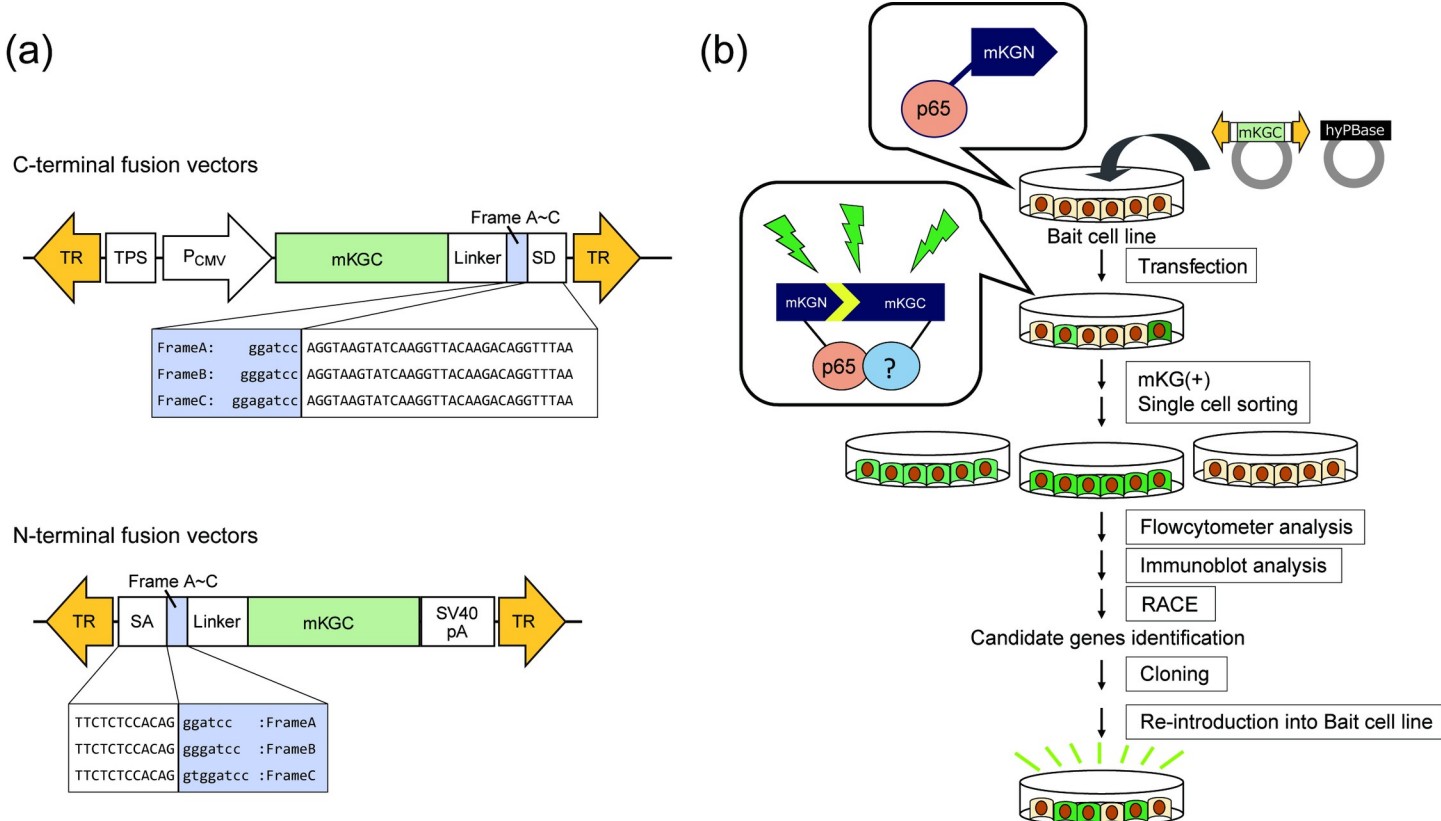

**Fig 1. Establishment of a screening system for p65 interacting proteins using a *PiggyBac* transposon system and BiFC.** (a) *PiggyBac* transposon donor vectors are designed to express fusion proteins with mKGC using a splicing mechanism. Three frames were prepared for each of the vectors to be fused to the C-terminal or N-terminal fragment of mKG. TR, terminal repeat; TPS, transcription pause site; SD, splicing donor; SA, splicing acceptor. (b) A transposon donor vector encoding mKGC (Fig 1A) was introduced with hyPBase vector into a bait cell line (p65-mKGN, HeLa). After 7 days, the mKG-positive population was recovered using FACS. After expansion, mKG-positive cells were cloned by single-cell sorting. Clones with high fluorescence were selected, and the expression of the mKGC fusion protein was confirmed by immunoblotting. Genes were identified by 5′- or 3′-RACE (rapid amplification of cDNA end). The cDNA of the fusion protein was cloned and then returned to the bait strain to examine whether BiFC fluorescence was reconstituted.

**Table 1. Composition of 3′-RACE PCR reaction.**

| Template cDNA | 0.3 μL |
|---|---|
| 2× PCR buffer | 12.5 μL |
| 2 mM dNTPs | 5.0 μL |
| 20 μM each primer mix | 1.5 μL |
| KOD FX neo 1U/μL | 0.5 μL |
| MQ | to 25 μL |

## RNA recovery and cloning of candidate genes

RNA was isolated from cells seeded on 6-cm dish using 1 mL of Isogen (Nippon Gene) according to the manufacturer's protocol. For cloning the mKGC fusion gene, cDNA was synthesized from 5 μg of extracted RNA using Superscript III First-Strand Synthesis System (Life Technologies). The expression vector of each mKGC fusion gene was prepared by in-fusion reaction of PCR fragments and/or a DNA fragment treated with a restriction enzyme. The primers or the restriction enzyme to obtain each DNA fragment were shown in S1 Table of S1 File.

## Identification of C-terminal fusion gene by 3′-RACE

cDNA was synthesized from 5 μg of RNA using 3′-Full RACE Core Set (Takara Bio) according to the manufacturer's protocol. As the reverse transcription primer, 'Oligo dT-3 sites Adapter Primer' attached to the product was used. Coding sequence linked to mKGC was amplified using KOD FX neo polymerase (TOYOBO) (Tables 1 and 2). The PCR primer sequence is shown in Table 3.

## Identification of N-terminal fusion gene by 5′-RACE

cDNA was synthesized from 5 μg of RNA using SuperScript III Reverse Transcriptase (Thermo Fisher) according to the manufacturer's protocol using mKGC gene specific primer 5′CTGTGGCTGATGTAG3′. Next, 2.5 U of RNaseH and 1 U of RNaseA were added to the reverse transcription solution and reacted at 37°C for 1 h to degrade RNA. The cDNA was purified using NucleoSpin Tissue (Takara) and eluted with 50 μL water. Ten microliters of the eluted cDNA was reacted in a tailing buffer (10 mM Tris-HCl (pH 8.4), 25 mM KCl, 1.5 mM MgCl$_2$, 0.2 mM dCTP, 7 U of terminal deoxynucleotidyl transferase (Takara)) at 37°C for 1 h to add polyC oligo-nucleotides to the 5′ end, followed by heat inactivation at 80°C for 3 min. Next, nested PCR was conducted to amplify the specific sequence of the mKGC fusion gene using KOD FX neo (TOYOBO) (Tables 4 and 5) using PCR primers shown in Table 6.

## Immunoblotting and immunoprecipitation

For analyses of protein expression, cells were collected in TNE buffer (10 mM Tris-HCl (pH7.8), 150 mM NaCl, 1 mM EDTA, 1% NP-40). Protein concentrations were determined

**Table 2. 3′-RACE PCR condition.**

| 94°C | 2 min | |
|---|---|---|
| 98°C | 10 s | ×30 |
| 60°C | 30 s | |
| 68°C | 5 min | |
| 68°C | 7 min | |
| 15°C | ∞ | |

**Table 3. Primers.**

| Primers used for 3′-RACE | 5′-CCACTACATCAGCCACAGGC-3′ |
|---|---|
| | 5′-CTGATCTAGAGGTACCGGATCC-3′ |

using the Pierce™ BCA Protein Assay Kit (Thermo Scientific). Collected protein lysates were mixed with SDS-PAGE loading buffer (0.15 M Tris-HCl (pH 6.8), 6% (w/v) SDS, 0.003% (w/v) Bromophenol blue, 30% (w/v) glycerol, 15% (w/v) β-mercaptoethanol), sonicated for 5 min and then boiled at 100°C for 5 min followed by SDS-PAGE and immunoblotting. For immunoprecipitation, cultured cells were washed once in PBS and collected with a scraper. After centrifugation, cells were lysed in TNE buffer for 10 min with occasional vortexing on ice. Cell lysates were centrifuged and the supernatants were precleared by incubation with protein-G Sepharose beads (GE healthcare). Cleared lysates were incubated at 4°C with antibodies for 100 min and then with the protein-G Sepharose beads for further 120min. Beads were collected by centrifugation and washed three times with TNF buffer. Protein lysates were mixed with SDS-PAGE loading buffer, and then boiled at 100°C for 10 min followed by SDS-PAGE and immunoblotting. Table 7 shows the antibodies and conditions used.

## Reporter assay

The Dual Luciferase Reporter Assay Kit (Promega) was used for the reporter assays. HeLa cells were seeded in a 12-well dish at $1.0 \times 10^5$ cells/well. The next morning, 1 μg of DNA 0.5 μg gene expression vector, 0.5 μg pGL4.32 (Promega) and 4 μg of PEI were mixed in 100 μL Opti-MEM and applied to the cells. Six hours later, the medium was replaced with culture medium. The next day, 20 ng/mL TNFα was added, and after the indicated time, cell lysates were recovered with 300 μL 1× Passive Lysis Buffer (Promega). Luciferase activity was measured using TriSTAR2S LB942 (BERTHOLD).

# Results and discussion

## Establishment of a screening system using *PiggyBac* transposon and BiFC

To establish a bait cell line, an N-terminal fragment (168 aa) of monomeric Kusabira-Green (mKG) was fused to a fragment of p65 (residues 190–291 of the full-length protein) and stably expressed in HeLa cells. The bait strain was cloned by FACS, and a clone was selected that had a high p65-mKGN expression and high fluorescence with the introduction of p50 (residues 247–352)-mKGC, a known binding partner. To fuse any fragment to mKGC, a total of 6 different *PiggyBac* transposon vectors (gene X is fused to C-terminal of mKG or N-terminal of mKG in each 3 frame-variations) were constructed (Fig 1A) and introduced into bait cells. Screening was performed by plating $8 \times 10^4$ cells/sample on a 12-well plate and introducing each of the six different vectors with the hyPBase transposase vector (Fig 1B). After 7 days, the mKG-positive cells were isolated by FACS. From 15 independent screenings, 9 different clones

**Table 4. Composition of 5′-RACE PCR.**

| | |
|---|---|
| Template cDNA | 5.0 μL |
| 2× PCR buffer | 25 μL |
| 2 mM dNTPs | 10 μL |
| 10 μM each primer mix | 2.0 μL |
| KOD FX neo 1 U/μL | 1.0 μL |
| MQ | to 50 μL |

**Table 5. 5′-RACE PCR condition.**

| 94˚C | 2 min | |
|---|---|---|
| 98˚C | 10 s | ×35 |
| 50˚C | 30 s | |
| 68˚C | 5 min | |
| 68˚C | 5 min | |
| 15˚C | ∞ | |

with high fluorescence compared to an empty control were isolated (Fig 2) and it was confirmed that the mKGC fusion protein was expressed in these cells (S1 Fig in S1 File).

In order to identify the C-terminal fusion gene in clones #1 to #5, 3′-RACE was performed revealing that exons from *CACYBP* (Clone#1&2), *HSP90AB1* (Clone #3), *PKM* (Clone#4), and *KRT8* (Clone #5) were fused with mKGC in frame (S2 Fig in S1 File). Similarly, the N-terminal fusion gene in clones #6 to #9 was analyzed by 5′-RACE and revealed that exons from *ANXA2* (Clone #6) and *HSPA8* (Clones #7, 8, 9) were fused to mKGC in frame (S2 Fig in S1 File).

Next, the candidate genes were cloned as mKGC fusions from each cell clone into the pPB-P$_{CMV}$-IRES-Puro$^R$ vector, introduced into the bait strain, and analyzed by flow cytometry. We confirmed that all samples were mKG-positive (Fig 3A). Among these, KRT8 (clone #5) had the weakest fluorescence (Figs 2 and 3A), and we did not perform further analyses with this clone. p50 was not identified as a binding partner in our screening, possibly because the transposon had low preference for the p50 locus, or, depending on the way in which the exon is chosen, not all of the trapped p50 peptide could cause BiFC, due to factors such as lack of required area or conformation. It is also possible that the total coverage of genes trapped by the vector was not high enough.

Immunoblot analyses (Fig 3B) revealed that the trapped mKGC fusion protein expressed in each clone and the proteins expressed from the isolated cDNAs were consistent for CACYBP, PKM, HSP90AB1, ANXA2, and HSPA8. Among these, PKM [15], ANXA2 [16], HSPA8 [17] and HSP90AB1 [18, 19] are already known as regulators of NF-κB, indicating that the screening system worked correctly.

## Regulation of NF-κB by CACYBP

Among the genes identified from the isolated clones, *CACYBP*, identified from clones #1 and #2, is the only one not previously known to interact with p65. p65 is known to be localized in the cytoplasm in an inactive state. We first analyzed the localization of BiFC fluorescence with a confocal microscope for clone #1 cells. While the majority of the signal was seen homogeneously in the cytoplasm, some formed unidentified speckles (Fig 4A). The nature of the speckles is currently unknown. To confirm the binding of endogenous p65 to CACYBP, immunoprecipitation assay was performed. As expected, endogenous CACYBP was detected in the coprecipitated complex of endogenous p65 in lysates from HeLa wild-type strain (WT) or clones # 1 and # 2 (Fig 4B and 4C). In addition, a band was detected at the expected

**Table 6. Primers.**

| Primers used for 5′-RACE 1$^{st}$ PCR | 5′-GGCCACGCGTCGACTAGTACGGGGGGGGGGGGGGGGG-3′ |
|---|---|
| | 5′-TGAACTGGCACTTGTGGTTG-3′ |
| Primers used for 5′-RACE 2$^{nd}$ PCR | 5′-GGCCACGCGTCGACTAGTAC-3′ |
| | 5′-TCCTGAACCACCACTACCAC-3′ |

**Table 7. Antibody list.**

| Antibody | Model number | Company |
|---|---|---|
| Anti-monomeric Kusabira Green C terminal fragment mouse mAb | M149-3M | MBL |
| Rabbit(DA1E) mAb IgG XP isotype control | #3900S | Cell signaling |
| Anti-p65 rabbit mAb (D14E12) | #8242S | Cell signaling |
| Anti-CACYBP rabbit mAb (D43G11) | #8225S | Cell signaling |

molecular weight of mKGC-CACYBP only for the lysates of clones # 1 and # 2 (Fig 4B). CACYBP is involved in cell growth and differentiation by controlling dephosphorylation and ubiquitination of target proteins [20], but there are no reports that it regulates NF-κB activity. We therefore examined the effects of CACYBP on NF-κB activation using a 3κB-Fluc reporter assay. We found that CACYBP overexpression significantly enhanced the fold change of NF-κB activity induced by TNFα (Fig 4D). This suggests that CACYBP is a previously unrecognized regulator of NF-κB. It is known that p65 translocates to the nucleus by stimulation with TNFα. In order to investigate whether the complex localization was changed by the stimulation of TNFα, confocal microscopic analysis was conducted. There were no changes in TNFα stimulation at 0, 15, 30, and 60 min, and at 0, 4, and 24 h for both fluorescence signals (S3 Fig in S1 File). This is consistent with the fact that p65-mKGN, which contains only part of p65 (aa residues 190–291), does not have the ability to translocate into the nucleus. This result also indicated that mKGC-CACYBP does not release binding to p65-mKGN upon TNFα stimulation.

In conclusion, we have established a transposon- and BiFC-based convenient screening system for identifying protein–protein interactions. This system identified various binding factors by random capture of the peptide of the endogenous gene, using a simple procedure, without the need to prepare a library. The method is straightforward and easy to implement, and could be readily applied to other proteins, to explore new binding partners.

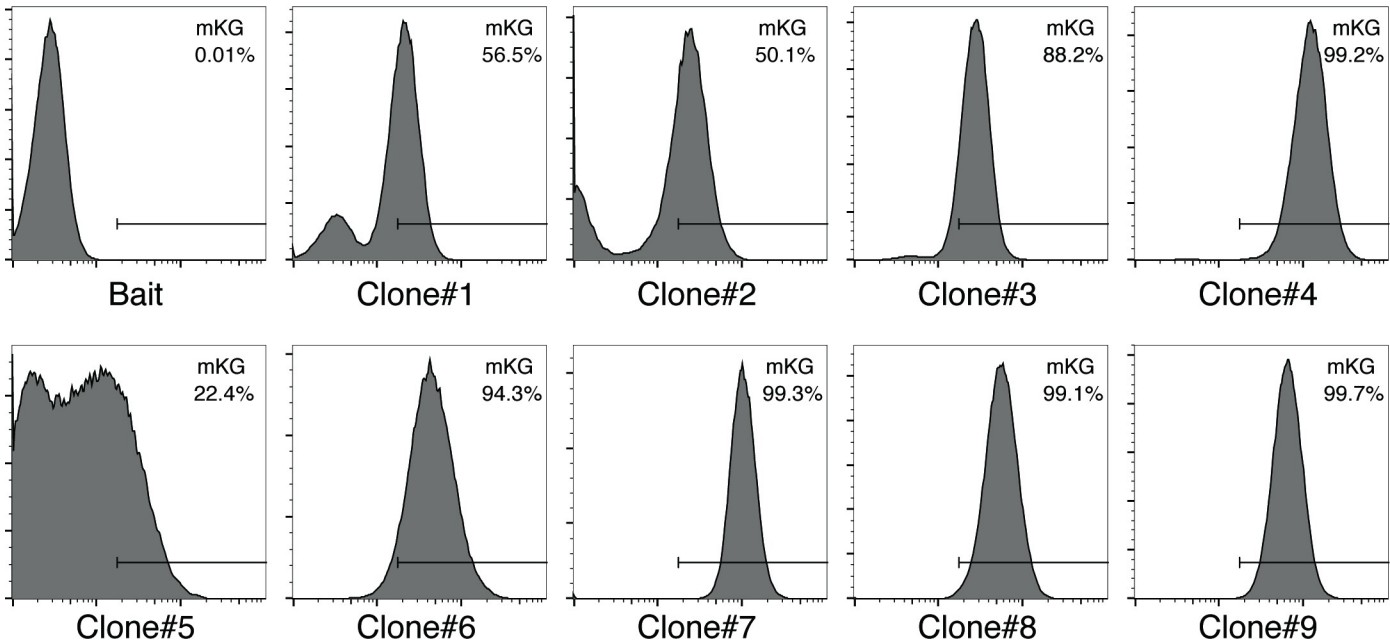

**Fig 2. Identification of positive clones by the BiFC screening system.** The fluorescence levels of reconstituted mKG by the BiFC mechanism within the cloned cells obtained by the screening were analyzed by flow cytometry.

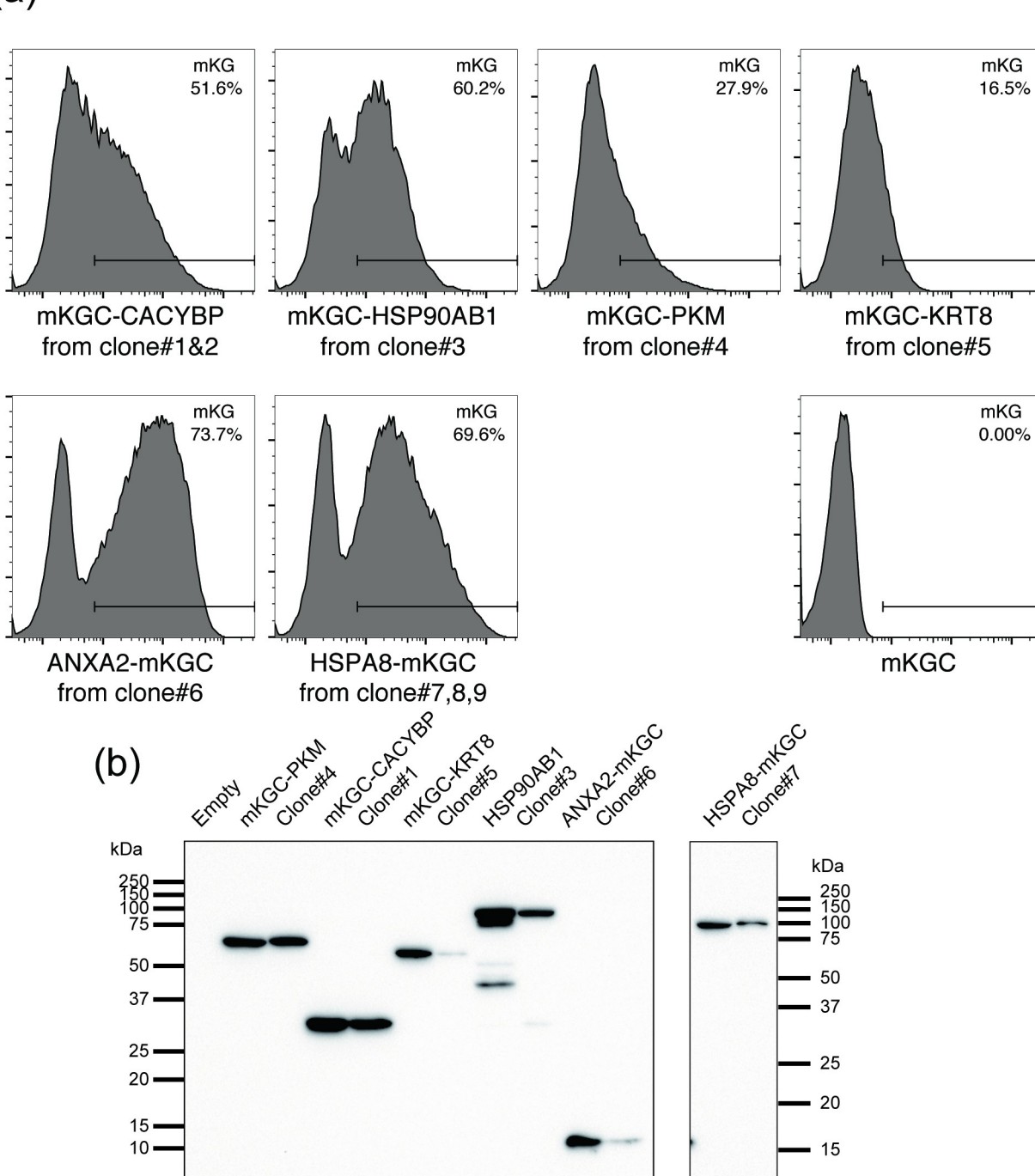

**Fig 3. Analyses of mKGC fusion genes identified by RACE.** (a) Each candidate fusion gene was cloned by PCR, incorporated into the pPB-P$_{CMV}$ vector and introduced into the bait strain (p65-mKGN). Clones were analyzed by flow cytometry to verify whether BiFC reconstitution occurred. (b) The lysates of the clones obtained from the screen and the lysate of the cell in which the cDNA of the mKGC fusion protein from the cloned cell was expressed were immunoblotted with an anti-mKGC antibody to compare their molecular weights.

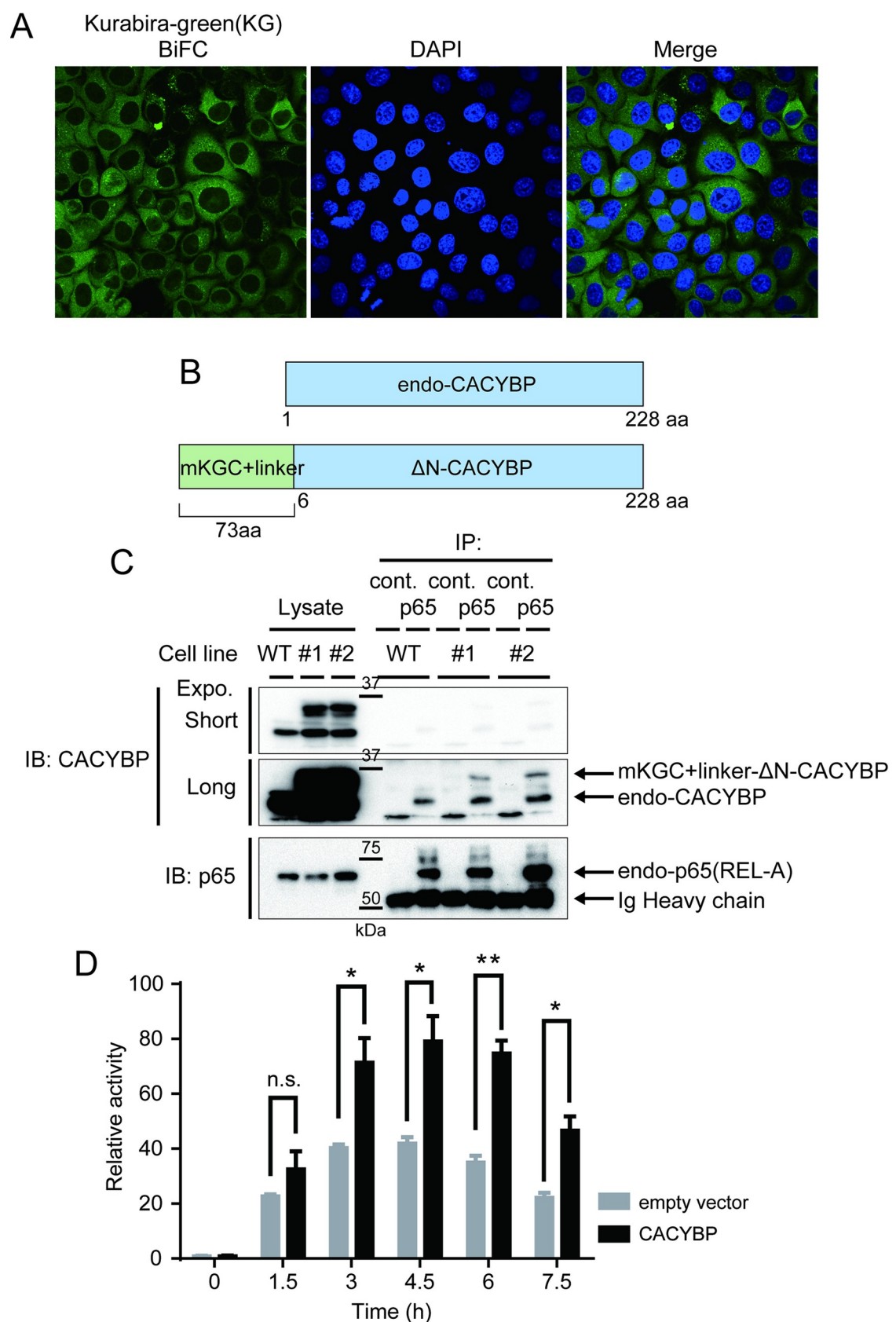

**Fig 4. Characterization of CACYBP.** A, Confocal microscopic analysis of BiFC fluorescence of clone #1 (Bait: p65-mKGN) where mKGC-CACYBP is expressed. DAPI (4′,6-diamidino-2-phenylindole) was used to stain the nucleus. B, Structures of endogenous CACYBP (upper) and CACYBP fused with mKGC (lower) in trapped clones #1 and #2. The latter was revealed by sequence analysis after cloning the cDNA of the fusion gene. C, Immunoprecipitation assay for binding between p65 and CACYBP. The antibody that recognizes endogenous p65 (but does not recognize p65-mKGN) was used for immunoprecipitation. IP, immunoprecipitation; IB, immunoblot; WT, HeLa wild type strain. D, Identification of NF-κB-regulating activity of CACYBP. A full-length CACYBP expression vector and a 3 kB-Fluc reporter vector were co-introduced into HeLa cells prior to addition of TNF-α. Cells were collected at each time shown and luciferase assays performed. Relative values were calculated with the mean value at 0 h as 1. A statistical processing t-test was performed (n = 3). $^{*}$, $p < 0.05$, $^{**}$, $p < 0.01$. This figure is representative of three independent experiments. n.s., not significant.

## Supporting information

**S1 File.**
(PDF)

## Acknowledgments

We are grateful to Drs. Allan Bradley and Kosuke Yusa (the Wellcome Trust Sanger Institute) for providing *piggyBac* transposon's backbone donor vectors [21] and a hyPBase vector [8]. We also thank Jun-ichiro Inoue for supporting NF-κB experiments, Jiro Fujimoto for extensive discussions, Kumiko Semba for the secretarial assistance, and Enago (https://www.enago.jp) for the English language review.

## Author Contributions

**Conceptualization:** Kosuke Ishikawa.

**Investigation:** Honami Miyakura, Mei Fukuda, Hiroya Enomoto.

**Methodology:** Kosuke Ishikawa.

**Resources:** Shinya Watanabe.

**Supervision:** Kosuke Ishikawa, Kentaro Semba.

**Validation:** Honami Miyakura.

**Writing – original draft:** Honami Miyakura, Kentaro Semba.

**Writing – review & editing:** Kosuke Ishikawa.

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
