## [Decision Letter · Decision Letter 0]

26 Aug 2020

PONE-D-20-15261

A new screening system for identifying interacting proteins using biomolecular fluorescence complementation

PLOS ONE

Dear Dr. Ishikawa,

Thank you for submitting your manuscript to PLOS ONE. After careful consideration, we feel that it has merit but does not fully meet PLOS ONE’s publication criteria as it currently stands. Therefore, we invite you to submit a revised version of the manuscript that addresses the points raised during the review process.

We look forward to receiving your revised manuscript.

Kind regards,

George Mosialos

Academic Editor

PLOS ONE

Journal Requirements:

2. We note that your Methods section is included in the supporting information file. Please include this methods section within the main manuscript file.

3. One of the noted authors is a group or consortium [Japan Biological Informatics Consortium (JBiC)]. In addition to naming the author group, please list the individual authors and affiliations within this group in the acknowledgments section of your manuscript. Please also indicate clearly a lead author for this group along with a contact email address.

Reviewers' comments:

Reviewer's Responses to Questions

**Comments to the Author**

1. Is the manuscript technically sound, and do the data support the conclusions?

Reviewer #1: Partly

Reviewer #2: Yes

2. Has the statistical analysis been performed appropriately and rigorously? 

Reviewer #1: N/A

Reviewer #2: I Don't Know

3. Have the authors made all data underlying the findings in their manuscript fully available?

Reviewer #1: Yes

Reviewer #2: Yes

4. Is the manuscript presented in an intelligible fashion and written in standard English?

Reviewer #1: Yes

Reviewer #2: Yes

5. Review Comments to the Author

Reviewer #1: The authors present a methodology based on bimolecular fluorescence complementation (BiFC) and validate it by analyzing interactors of the p65 subunit of NF-kappaB. While this approach may prove to be an interesting addition to existing methodologies to study protein-protein interactions, the manuscript needs strengthening in order to be more convincing about its general applicability and usefulness.

Major points:

1. The authors often mention their method as being novel, but is just a different implementation of an established procedure (BiFC), by using different fluorescent molecules. It should be more clearly stated in the manuscript that this is a variation of an existing method.

2. The introductory remarks about FRET are not entirely correct. FRET intensity depends on various factors, such as orientation of the molecules and proximity of the molecules, so over-expression in not necessarily required. FRET might not be able to detect weak/transient interactions, but in contrast to BiFC it can probe the dynamics of a system. In general, limitations of BiFC should be properly introduced together with its strengths, such as the time required for the complex to fold and fluoresce (it is NOT a real-time approach), the fact that it is irreversible (cumulative signal - hence no real time information), and that negative controls are not so straight forward.

3. In discussing the lack of interaction of p65 with p50, the authors suggest that the trapped p50 peptide may not conformationally cause BiFC. However, they mention early in the results that they had a high fluorescence signal with the introduction of p50 (residues 247–352)-mKGC. This needs to be clarified further by tagging whole p50 and seeing if they get complementation upon transfection together with tagged p65. How does this compare with other BiFC systems? Previous work (Hu et al. 2002) has shown an interaction between the two using BiFC with different fluorescent tags. There is no further investigation or discussion of these discrepancies.

4. The overall impression from this manuscript is that it needs strengthening, either in the characterization of the novel p65 interactor that was identified (very little is done in this respect) or in the direct comparison with existing BiFC approaches, to reveal any advantages of the methodology presented here (what is the rational for using the piggyBac system over others, is the folding time faster, does it give a stronger signal, can it be used for proteins present in not easily accessible subcellular compartments, etc).

Minor point

In the abstract, 5 p65 interactors are reported, while 6 are shown in the results.

Reviewer #2: Miyakura H et al. reported a new screening system by combining BiFC and transposon-mediated library creation. The authors found CACYBP as a novel binding partner of p65. The screening system that the authors established is interesting and useful to identify protein-protein interaction in vivo. This manuscript would have broad interest to readers of PLoS One. This reviewer lists a few comments below to further strengthen the manuscript.

1. The authors should confirm the interaction between p65 and CACYBP by a method other than BiFC such as immunoprecipitation.

2. It will be informative to present subcellular localization of CACYBP along with p65 before and after TNF stimulation. In addition, where is mKG fluorescence observed, in nucleus, cytosol, or both?

3. Figure legends. The authors should describe more detailed information in order for readers to easily understand figures. For instance, abbreviations used in figures should be defined in figure legends.

6. PLOS authors have the option to publish the peer review history of their article (what does this mean?). If published, this will include your full peer review and any attached files.

Reviewer #1: No

Reviewer #2: No

---

## [Author Response · Author response to Decision Letter 0]

25 Mar 2021

> Journal Requirements:

> 

> When submitting your revision, we need you to address these additional requirements.

> 

> 

> 

> 1. Please ensure that your manuscript meets PLOS ONE's style requirements, including those for file naming. The PLOS ONE style templates can be found at

> 

> https://journals.plos.org/plosone/s/file?id=wjVg/PLOSOne_formatting_sample_main_body.pdf and

> 

> https://journals.plos.org/plosone/s/file?id=ba62/PLOSOne_formatting_sample_title_authors_affiliations.pdf

> 

I have revised the manuscript to meet the specified journal format.

> 

> 2. We note that your Methods section is included in the supporting information file. Please include this methods section within the main manuscript file.

> 

As requested, the Methods section was moved to the main manuscript file.

> 

> 3. One of the noted authors is a group or consortium [Japan Biological Informatics Consortium (JBiC)]. In addition to naming the author group, please list the individual authors and affiliations within this group in the acknowledgments section of your manuscript. Please also indicate clearly a lead author for this group along with a contact email address.

> 

The main author belongs to one of the laboratories within the Consortium. I have changed the citation to reflect this situation, and make the authorship clearer. 

> 

> 4. We note that you have included the phrase “data not shown” in your manuscript. Unfortunately, this does not meet our data sharing requirements. PLOS does not permit references to inaccessible data. We require that authors provide all relevant data within the paper, Supporting Information files, or in an acceptable, public repository. Please add a citation to support this phrase or upload the data that corresponds with these findings to a stable repository (such as Figshare or Dryad) and provide and URLs, DOIs, or accession numbers that may be used to access these data. Or, if the data are not a core part of the research being presented in your study, we ask that you remove the phrase that refers to these data.

The data were not a core part of the research, so we removed the phrase from the manuscript.

> 5. Review Comments to the Author

> Reviewer #1: The authors present a methodology based on bimolecular fluorescence complementation (BiFC) and validate it by analyzing interactors of the p65 subunit of NF-kappaB. While this approach may prove to be an interesting addition to existing methodologies to study protein-protein interactions, the manuscript needs strengthening in order to be more convincing about its general applicability and usefulness.

> 

> Major points:

> 

> 1. The authors often mention their method as being novel, but is just a different implementation of an established procedure (BiFC), by using different fluorescent molecules. It should be more clearly stated in the manuscript that this is a variation of an existing method.

> 

Thank you for your suggestions. To avoid misunderstanding, we removed "new" from the title. The full text of the title was changed to “A screening system for identifying interacting proteins using biomolecular fluorescence complementation and transposon gene trap’. The words ‘novel method’ in abstract was replaced with ‘valuable method”. To avoid giving the impression that the BiFC method itself was developed in this work, we replaced the statement “We have established a new screening system for identifying interacting proteins using biomolecular fluorescence complementation (BiFC).” in abstract with the statement “We have established a new screening system for identifying interacting proteins by combining biomolecular fluorescence complementation (BiFC) and a transposon gene trap system”

> 2. The introductory remarks about FRET are not entirely correct. FRET intensity depends on various factors, such as orientation of the molecules and proximity of the molecules, so over-expression in not necessarily required. FRET might not be able to detect weak/transient interactions, but in contrast to BiFC it can probe the dynamics of a system. In general, limitations of BiFC should be properly introduced together with its strengths, such as the time required for the complex to fold and fluoresce (it is NOT a real-time approach), the fact that it is irreversible (cumulative signal - hence no real time information), and that negative controls are not so straight forward.

We appreciate your comments. In response to this advice, the description “Fluorescence resonance energy transfer (FRET) can be used to visualize protein interactions in living cells, but high levels of expression are required otherwise FRET cannot be detected even when proteins bind.” was changed to “Fluorescence resonance energy transfer (FRET) can be used to probe the dynamics of protein interactions in living cells. However, the visualization is real-time, the fluorescence does not accumulate, and requires the use of highly sensitive detection devices.” The description “The reconstituted fluorescent protein is very stable, allowing for detection of even weak interactions.” was changed to the statement “Although it may take longer for fluorescence to mature than in the case of the FRET system, the fluorescence accumulates and becomes stable, allowing for the detection of even weak interactions.”

> 

> 3. In discussing the lack of interaction of p65 with p50, the authors suggest that the trapped p50 peptide may not conformationally cause BiFC. However, they mention early in the results that they had a high fluorescence signal with the introduction of p50 (residues 247–352)-mKGC. This needs to be clarified further by tagging whole p50 and seeing if they get complementation upon transfection together with tagged p65. How does this compare with other BiFC systems? Previous work (Hu et al. 2002) has shown an interaction between the two using BiFC with different fluorescent tags. There is no further investigation or discussion of these discrepancies.

Thank you for your detailed review. I corrected the part that was lacking in the statements. In this discussion, we described possible reasons why p50 was not identified. One possibility is that we have not obtained a p50 fragment with a suitable binding region at which BiFC can occur, such as residues 247–352 as well as other constructs, including those suggested by the reviewer (Hu et al.)). It may depend on how the exon is chosen upon gene trapping by the trap vector. The assumption that not all trapped fragments cause BiFC may not be wrong, due to differences in factors such as folding and conformation, required for binding, and there are no discrepancies. It is also possible that the total coverage of genes trapped by the vector was not high enough, in which case not all known p65-interacting proteins would be detected using this method. However, in this study, PKM, HSP90AB1, ANXA2, and HSPA8, which are known to bind to p65, were identified using this method. The relevant description has been corrected as follows: “p50 was not identified as a binding partner in our screening, possibly because the transposon had low preference for the p50 locus, or, depending on the way in which the exon is chosen, not all of the trapped p50 peptide could cause BiFC, due to factors such as lack of required area or conformation. It is also possible that the total coverage of genes trapped by the vector was not high enough.”

>

> 4. The overall impression from this manuscript is that it needs strengthening, either in the characterization of the novel p65 interactor that was identified (very little is done in this respect) or in the direct comparison with existing BiFC approaches, to reveal any advantages of the methodology presented here (what is the rational for using the piggyBac system over others, is the folding time faster, does it give a stronger signal, can it be used for proteins present in not easily accessible subcellular compartments, etc).

Thank you for your helpful suggestions. We do not think it is necessary to compare the methodology presented here with the conventional BiFC method, which is often used for proteins that have a fixed target to be analyzed. However, we utilized BiFC theory in combination with the properties of transposons to create a convenient screening system for searching for binding proteins. The transposons explore the genome in depth, and any protein fragments are fused randomly. This approach is therefore more straightforward to apply than normal library creation methods. These points were made in the previous submission. In this revised version, the conclusions section in the ‘Results and discussion’ was corrected and the major advantages were added as follows: “This system identifies various binding factors by random capture of the peptide of the endogenous gene, using a simple procedure, without the need to prepare a library. The method is straightforward and easy to implement, and could be readily applied to other proteins, to explore new binding partners.” Using this system we identified a new binding factor (CACYBP) for p65, without significant effort, and showed its function in enhancing the activation of NF-kB by TNFα. In a new experiment examining the binding of CACYBP to p65, added in this revision, the interaction between the endogenous proteins was confirmed by immunoprecipitation assays using p65 and CACYBP antibodies (Fig4 B and C). Also, as suggested by another reviewer, we added fluorescence image data from a confocal microscope (Fig4A, and S3 Figure). By adding these results to this revised edition, we believe that this method can be emphasized to be both simple and useful. Since the method is highly versatile, it can be readily applied to other proteins to explore new binding partners. The approach therefore has considerable potential to be valuable in future research.

>

> Minor point

> 

> In the abstract, 5 p65 interactors are reported, while 6 are shown in the results.

> 

Thank you for your detailed review. As described in the previous submission, KRT8 was weakest fluorescent signal detected by BiFC (Fig3A), and therefore should not necessarily be presented. However, its lysate was used in the immunoblot (Fig3B). Its description could be completely eliminated from the manuscript, but since it was included in the immunoblot image in Fig3B, we decided it needed to be described.

> Reviewer #2: Miyakura H et al. reported a new screening system by combining BiFC and transposon-mediated library creation. The authors found CACYBP as a novel binding partner of p65. The screening system that the authors established is interesting and useful to identify protein-protein interaction in vivo. This manuscript would have broad interest to readers of PLoS One. This reviewer lists a few comments below to further strengthen the manuscript.

> 

> 1. The authors should confirm the interaction between p65 and CACYBP by a method other than BiFC such as immunoprecipitation.

Thank you for raising this issue. New immunoprecipitation experiments were conducted, and binding between endogenous p65 and either endo-CACYBP or mKGC-CACYBP was confirmed. These results are included in this revised edition (Fig 4B and C).

> 2. It will be informative to present subcellular localization of CACYBP along with p65 before and after TNF stimulation. In addition, where is mKG fluorescence observed, in nucleus, cytosol, or both?

Thank you for your suggestion. The localization was newly analyzed using a confocal microscope (Fig 4A). A signal was observed in the cytoplasmic region outside the nucleus. While the majority of the signal was seen homogeneously in the cytoplasm, some signal formed unidentified speckles. The nature of this speckle is currently not known. This pattern did not seem to change during the 0-24h period of TNFα stimulation. In this revised version, these data are shown in Fig 4A and S3 figures.

> 3. Figure legends. The authors should describe more detailed information in order for readers to easily understand figures. For instance, abbreviations used in figures should be defined in figure legends.

Thank you for your helpful comment. We have revised the relevant legends in the manuscript, according to this advice.

---

## [Decision Letter · Decision Letter 1]

23 Apr 2021

A screening system for identifying interacting proteins using biomolecular fluorescence complementation and transposon gene trap

PONE-D-20-15261R1

Dear Dr. Ishikawa,

We’re pleased to inform you that your manuscript has been judged scientifically suitable for publication and will be formally accepted for publication once it meets all outstanding technical requirements.

Kind regards,

George Mosialos

Academic Editor

PLOS ONE

Reviewers' comments:

Reviewer's Responses to Questions

**Comments to the Author**

1. If the authors have adequately addressed your comments raised in a previous round of review and you feel that this manuscript is now acceptable for publication, you may indicate that here to bypass the “Comments to the Author” section, enter your conflict of interest statement in the “Confidential to Editor” section, and submit your "Accept" recommendation.

Reviewer #1: All comments have been addressed

Reviewer #2: All comments have been addressed

2. Is the manuscript technically sound, and do the data support the conclusions?

Reviewer #1: Yes

Reviewer #2: Yes

3. Has the statistical analysis been performed appropriately and rigorously? 

Reviewer #1: Yes

Reviewer #2: Yes

4. Have the authors made all data underlying the findings in their manuscript fully available?

Reviewer #1: Yes

Reviewer #2: Yes

5. Is the manuscript presented in an intelligible fashion and written in standard English?

Reviewer #1: Yes

Reviewer #2: Yes

6. Review Comments to the Author

Reviewer #1: (No Response)

Reviewer #2: The reviewers have responded to the comments raised by the reviewer. Now the manuscript will be suitable for publication in PLoS ONE.

7. PLOS authors have the option to publish the peer review history of their article (what does this mean?). If published, this will include your full peer review and any attached files.

Reviewer #1: No

Reviewer #2: **Yes: **Hiroyasu Nakano

---

## [Editor Report · Acceptance letter]

27 Apr 2021

PONE-D-20-15261R1 

A screening system for identifying interacting proteins using biomolecular fluorescence complementation and transposon gene trap 

Dear Dr. Ishikawa:

I'm pleased to inform you that your manuscript has been deemed suitable for publication in PLOS ONE. Congratulations! Your manuscript is now with our production department. 

Kind regards, 

on behalf of

Dr. George Mosialos 

Academic Editor

PLOS ONE